# Knowledge-inspired 3D Scene Graph Prediction in Point Cloud

**Shoulong Zhang**
Beihang University
shoulong.zhang@buaa.edu.cn

**Shuai Li** *
Beihang University
Peng Cheng Laboratory
lishuai@buaa.edu.cn

**Aimin Hao**
Beihang University
Peng Cheng Laboratory
ham@buaa.edu.cn

**Hong Qin** *
Stony Brook University (SUNY)
qin@cs.stonybrook.edu

## Abstract

Prior knowledge integration helps identify semantic entities and their relationships in a graphical representation, however, its meaningful abstraction and intervention remain elusive. This paper advocates a knowledge-inspired 3D scene graph prediction method solely based on point clouds. At the mathematical modeling level, we formulate the task as two sub-problems: knowledge learning and scene graph prediction with learned prior knowledge. Unlike conventional methods that learn knowledge embedding and regular patterns from encoded visual information, we propose to suppress the misunderstandings caused by appearance similarities and other perceptual confusion. At the network design level, we devise a graph auto-encoder to automatically extract class-dependent representations and topological patterns from the one-hot class labels and their intrinsic graphical structures, so that the prior knowledge can avoid perceptual errors and noises. We further devise a scene graph prediction model to predict credible relationship triplets by incorporating the related prototype knowledge with perceptual information. Comprehensive experiments confirm that, our method can successfully learn representative knowledge embedding, and the obtained prior knowledge can effectively enhance the accuracy of relationship predictions. Our thorough evaluations indicate the new method can achieve the state-of-the-art performance compared with other scene graph prediction methods.

## 1 Introduction and Motivation

Scene graph has been studied as a viable means towards better interpretation of scene context, which encodes the semantic elements and their complex relationships of a scene [16, 32, 33, 11, 31, 9]. As an enriched scene structure representation, scene graph parsing from 3D data has recently interested researchers in 3D vision for the tasks of scene understanding [2], VR/AR scene interactions [25], and robot navigation [7]. However, the scanned 3D data has a intrinsic nature of incompleteness [1, 24], geometrical similarity across categories, and other visual challenges. These perception errors result in extra difficulties in object and relationship recognition purely based on visual inputs.

Fortunately, in cognitive psychology, besides visual perception, human beings also associate objects and concepts with certain abstract symbolic class-dependent representations and regular combinations

---

*Corresponding authors.

35th Conference on Neural Information Processing Systems (NeurIPS 2021).

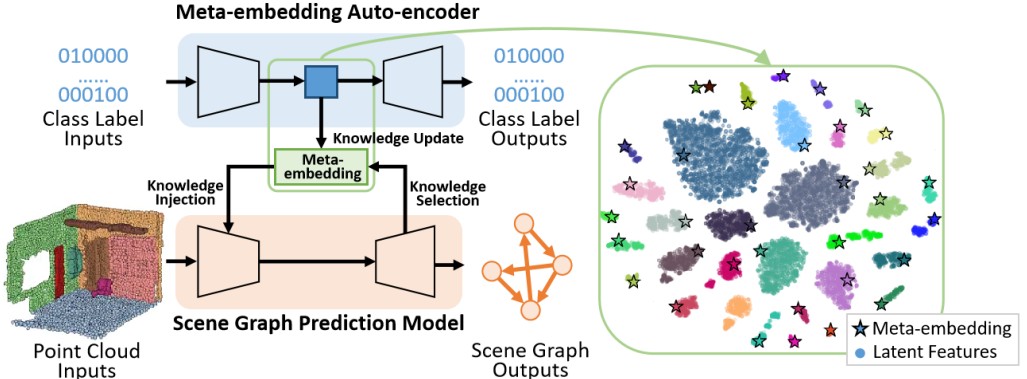

Figure 1: Overview of the knowledge-inspired 3D scene graph prediction method. We decompose the problem into two sub-tasks. First, we learn the meta-embedding only from the class labels with their graphical structures avoiding perceptual errors. The figure on the right shows that our meta-embedding (stars) can effectively represent the latent class features (points) in the latent space. Then, our scene graph prediction model selects related meta-embedding as prior knowledge to classify object entities and their relationships with point cloud perceptual features.

of scene context, known as knowledge. However, utilizing prior knowledge still remains challenging in scene graph prediction. Prior knowledge-incorporated methods have several limitations. Static knowledge (e.g., heuristic co-occurrence statistics [37], mined triplet facts [8], or knowledge graph [35]) could be incomplete and inaccurate depending on the quality of knowledge sources and the domain knowledge covered. Thus, several methods tend to use learning-based knowledge to capture the graphical structure and regular combinations in the exact domain. However, the knowledge learned from visual information also inherits perceptual confusion into the knowledge embedding and network parameters [22, 36].

To ameliorate, this paper advocates a novel 3D scene graph prediction method with learned prior knowledge. The knowledge can be encoded in various forms (e.g., vectors, graphs, and natural language). In our work, we choose class-dependent prototypical memories as the carrier of knowledge to tackle the key challenges mentioned above. We re-formulate the target problem into two sub-tasks: knowledge learning and knowledge-inspired scene graph prediction. Unlike previous works, we learn the knowledge only depending on semantic categories and regular structural patterns without any visual confusion. The knowledge learning model is a graph auto-encoder that takes the one-hot class labels as inputs with scene graph structure annotations. Since we do not involve any geometric and visual features, the latent embedding indeed encodes only category-related regular structure patterns. To correctly record this prior knowledge, we enforce a group of prototypes [23], namely meta-embedding, to approach the latent areas of the corresponding category in the metric space. As we illustrated in Figure 1, the meta-embedding is capable of representing each semantic class as prototypical knowledge. Once trained, the scene graph prediction model unites the geometric features and the corresponding knowledge meta-embedding to predict each element in a triplet with the <*subject, predicate, object*> structure. Extensive experiments confirm that the effectiveness of knowledge extraction and the injection of the meta-embedding can improve the scene graph prediction accuracy.

The **primary contributions** of this paper could be summarized as follows: (1) We introduce a novel knowledge-inspired scene graph prediction method that directly functions on 3D scene point clouds. To achieve our goal, we re-formulate the problem into two sub-tasks: knowledge learning and scene graph prediction with knowledge intervention; (2) We propose a graph auto-encoder, called meta-embedding, to learn a group of class-dependent prototypical representations. The learned class prototypes can avoid perceptual errors and effectively encode prior structural knowledge in a more distinguishable latent space; and (3) We design a scene graph prediction model benefiting from the prior learned knowledge in order to predict credible relation triplets. With our learned knowledge, our scene graph prediction model can achieve state-of-the-art performance on the 3D semantic scene graph (3DSSG) dataset [29] compared to currently available methods.

## 2   Related Work

**Knowledge-based Scene Graph Classification.** Prior knowledge has been proven as an effective source of information to enhance object and relation recognition. Early methods use statistical co-occurrence as extra knowledge for scene graph inference. Zellers et al. observed that the predicate distribution is strongly dependent on the head and tail categories. It introduces a pre-computed bias into the final prediction [37]. The biased results naturally improve the most frequent combinations. To improve the classification performance on the rarely sampled ones, knowledge-embedded routing network (KERN) [4] implicitly fuses co-occurrence probabilities as weights in the message pass step to neutralize the long-tail phenomenon. Nevertheless, this integrated knowledge can hardly be used and studied in isolation since it is hard-coded as intrinsic parameters. Besides, some methods bring external knowledge bases into scene graph inference. Gu et al. extracted knowledge triplets from the ConceptNet knowledge bases [8]. Graph bridging network (GB-Net) adopts auxiliary edges as bridges that enable and facilitate message passing between knowledge graphs and scene graph [35]. This hand-crafted external information is highly dependent on the organization of knowledge and the quality of sources. Furthermore, several methods try to incorporate learning-based commonsense into scene graph inference. Zareian et al. proposed a local-global graph transformer to complete a masked scene graph by embedding the regular patterns into the network parameters [36]. More recently, Sharifzadeh et al. encoded inductive biases into the class-level prototypical representations learned from all previous perceptual outputs [22]. Inspired by [22], our method also adopts prototypical representations as prior knowledge. Nevertheless, our method focuses on class-dependent and graph structure information, which aims to combat the confusion resulting from perceptual errors.

**3D Scene Graph Prediction.** 3D scene graph has only recently been studied in the 3D vision community. There are mainly two approaches for 3D scene graph construction. The first group of works benefits from image-based scene graph techniques. Armeni et al. firstly introduced a 3D scene graph construction framework with 3D meshes and registered RGB panoramic images [2]. It adopts multi-view consistency to enforce the accuracy of object identification by 2D-3D projection. Furthermore, succeeding methods [13, 20] execute image-based scene graph prediction within dynamic simultaneously localization and mapping (SLAM) [12, 17, 30]. Kim et al. proposed a framework that merges the local graphs into a global one by fusing the same nodes in different images [13]. Spatial PerceptIon eNgine (SPIN) distinguishes human and object nodes by dynamic masking [20]. The second group of approaches directly process 3D data. Recently, Wald et al. proposed an end-to-end network, scene graph prediction network (SPGN) [29], which takes scene scans as input. Along with the network, they also published the 3DSSG dataset of reconstructed scene scans. Similar to this work, we investigate a novel model to extend the scene graph prediction methods using 3D scene data solely based on point clouds.

## 3   New Method

**Problem Formulation.** We define a scene graph $\mathcal{G}$ as a directed graph with sets of nodes and edges, $\mathcal{G} = \{\mathcal{V}, \mathcal{E}\}$. The node set $\mathcal{V}$ includes all object instances of a scene, and the edge set $\mathcal{E}$ consists of all possible connections of nodes as predicates in a relation triplet <*subject, predicate, object*>. The edges are directed by connecting neighbor nodes from subject to object. Unlike previous knowledge-embedded approaches [37, 4, 36], we formulate the knowledge-inspired scene graph prediction task as a multi-task problem: we intend to learn knowledge and scene graphs from 3D data simultaneously. Mathematically, given a 3D scene data $D$ with scene graph annotations, the training process is aimed to predict the probability of $P(\mathcal{G}, K|D)$, where $K$ is the prior knowledge. Furthermore, we can decompose the joint probability into two individual components:

$$P(\mathcal{G}, K|D) = P(K|D)P(\mathcal{G}|D, K). \tag{1}$$

This decomposition breaks the problem into two sub-tasks. The first sub-task, the predication of $P(K|D)$, extracts knowledge from existing 3D scene configurations. In this paper, the learned knowledge is in the form of prototypical embedding of each object and predicate category, named meta-embedding $K = \{\mathbf{e}_{meta}\}$. The set of meta-embedding is expected to represent each semantic class and also encode the corresponding class's regular patterns in scene graphs. Towards this goal, we design a graph auto-encoder network, which only takes the one-hot labels as input without any perceptual information. The graphical encoder also can extract the topological interactions of each class. We close the distance of latent outputs and the related meta-embedding in metric space so that the meta-embedding records the prior knowledge of the training data.

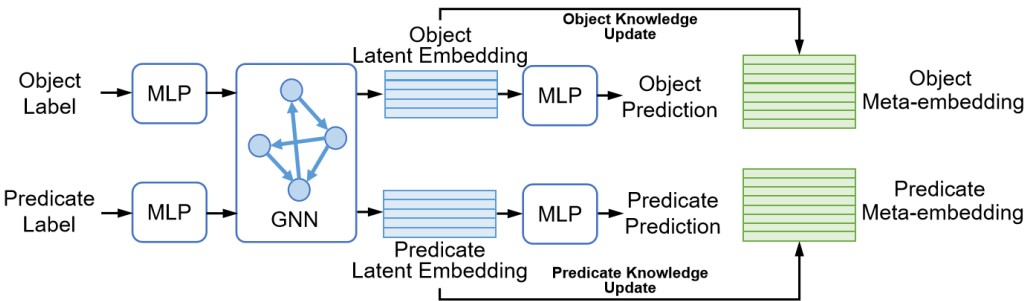

Figure 2: Overview of the meta-embedding learning auto-encoder.

The second sub-task, $P(\mathcal{G}|D, K)$, incorporates the meta-embedding and perceptual features to predict the possible scene graph elements. It is also the expected mathematical model in the evaluation stage with unseen 3D scene data.

**Meta-embedding and Learning Model.** This model is designed to learn class-dependent prototypical knowledge embedding with a graph auto-encoder. The overview of the meta-embedding learning model is illustrated in Figure 2. Since we consider the class-related patterns instead of perceptual information, the auto-encoder only takes the one-hot class labels as input. The first two multi-layer perceptron (MLP) blocks lift one-hot vectors into latent space with the same dimension size. Then, we assume that each object pair can have a relation (including none relation) and fully connect them as a graph where predicates are represented as edges. We design a graph neural network to further encode the graphical patterns of each node and edge in scene graphs by message passing [6]. At the network design level, we adapt the message passing mechanism to the directed graph structure. To calculate the latent representations $\{\mathbf{h}_v^{n,l}\}$ for all nodes $v \in \mathcal{V}$ and $\{\mathbf{h}_{uv}^{e,l}\}$ for all directed edges $(u, v) \in \mathcal{E}$ in every layer $l$, the message passing of nodes and edges can be formulated as:

$$\mathbf{h}_v^{n,l+1} = \text{GRU}^{n,l}(\mathbf{h}_v^{n,l}, m^{n,l}(v)), \tag{2}$$

$$\mathbf{h}_{uv}^{e,l+1} = \text{GRU}^{e,l}(\mathbf{h}_{uv}^{e,l}, m^{e,l}(u, v)), \tag{3}$$

where $m^{n,l}$ and $m^{e,l}$ are the parameterized neural networks that calculate messages in layer $l$. We adopt the gated recurrent unit (GRU) [5] to update the hidden state for reserving initial information. It is worth noting that a node could be a subject or object in a relation triplet. We consider both conditions in message computation. For node $v$ and one of its neighbors $u$, the message as subject $m^{s,l}(v)$ and as object $m^{o,l}(v)$ are computed by:

$$m^{s,l}(v) = LN(\phi_p(\mathbf{h}_{vu}^{e,l}) + \phi_o(\mathbf{h}_u^{e,l})), \tag{4}$$

$$m^{o,l}(v) = LN(\phi_p(\mathbf{h}_{uv}^{e,l}) + \phi_s(\mathbf{h}_u^{e,l})), \tag{5}$$

where $\phi_s$, $\phi_p$, and $\phi_o$ are three non-linear transformations for subject, predicate, and object respectively. To stabilize the numerical calculation in recurrent mechanism, we adopt the layer normalization ($LN$) [3] after feature fusion. The final message of node $v$ is the average of $m_v^{s,l}$ and $m_v^{o,l}$:

$$m^{n,l}(v) = \frac{1}{|\mathcal{R}_v^s| + |\mathcal{R}_v^o|} \left( \sum_{u \in \mathcal{R}_v^s} (m^{s,l}(u)) + \sum_{u \in \mathcal{R}_v^o} (m^{o,l}(u)) \right), \tag{6}$$

where $|\cdot|$ denotes cardinality and $\mathcal{R}_v^s$ and $\mathcal{R}_v^o$ are the set of connections of node $v$ as subject and as object respectively. The calculation of edge $(u, v)$ message is:

$$m^{e,l}(u, v) = LN(\phi_s(\mathbf{h}_u^{n,l}) + \phi_o(\mathbf{h}_v^{n,l})). \tag{7}$$

We use the sum pooling operation to the stacked hidden states as the output model of the graph encoder. Since we know the class labels of the latent output embedding of the graph encoder, we update the corresponding meta-embedding vectors by reducing their distances (e.g., Euclidean distance) in metric space. Given the latent embedding of nodes and those of edges, we select the corresponding class vectors from the object meta-embedding and the predicate meta-embedding. Then we reduce the distance between those embeddings as a loss function. As the values of

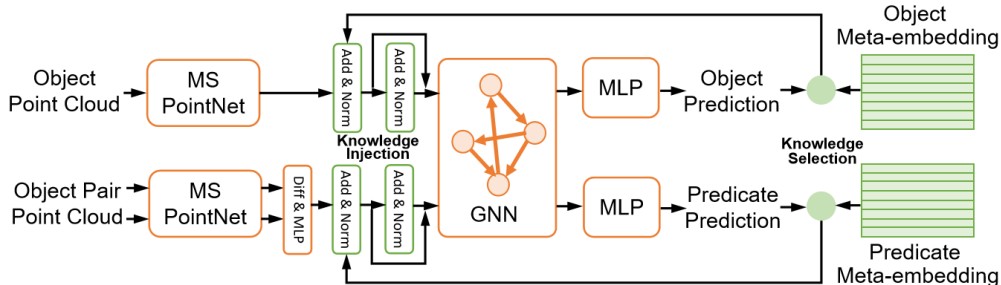

Figure 3: Overview of the scene graph prediction model with the learned meta-embedding.

meta-embedding are initialized as network parameters, they can eventually be updated with back-propagation [21]. The meta-embedding is updated only when seeing the related category, not depending on the number of training samples. This property makes the meta-embeddings qualified to be prototypical representations for even less frequent categories. The last two MLPs predict one-hot labels to enforce the latent embedding is class-related and complete the auto-encoder structure.

**Knowledge-inspired Scene Graph Prediction Model.** This model is aimed to fuse prior meta-embedding into the scene graph prediction task. Figure 3 shows the pipeline of the knowledge-inspired scene graph prediction model. Taking a scene point cloud with instance masks as input, we expect to encode geometrical features from both objects and object pairs. Based on the pioneering point set encoder, PointNet [18], we enhance the original structure to capture multi-scale geometric information like several point completion methods [10, 34]. For the object nodes, we directly use our modified multi-scale PointNet (MS PointNet) on the normalized point clouds. We first encode subject and object point sets separately for the object-pair edges and concatenate the latent features with the center coordinates of two point sets. To better encode the contradictory relationships (e.g., left and right, lower than and higher than), we adopt subtraction as an anti-symmetric operation on these two features. After a transformation by a two-layer MLP, we obtain the initial edge embedding from the object pairs.

We adopt the same graph neural network design in the meta-embedding learning stage to capture the global context with perceptual information. In the first iteration, the graph neural network only takes the geometric features of the PointNet as input. The last two MLPs output the node and edge classification possibilities and offer the clues to select meta-embedding vectors of related categories. We select the five most confident meta-embedding for each node and edge and fuse them with the perception features. Based on our experiments, the large number of selected embedding is inevitable to introduce wrong knowledge, while a small meta-embedding set might contain no proper knowledge. In practice, we choose five meta-embedding as a compromise. In the second iteration, since the perceptual embedding and the knowledge are in different latent spaces, we transform them by two-layer feed-forward networks $f$ and $g$ respectively before directly fusing them. Denoting by $\{\mathbf{e}^i_{meta}\}_{i=1...k}$ the selected meta-embedding and $\mathbf{x}$ the features encoded by multi-scale PointNet, we fuse the knowledge and perceptual information similar as in [22]:

$$\mathbf{z} = LN(f(\mathbf{x}) + \sum_{i=1}^{k} g(\mathbf{e}^i_{meta})), \tag{8}$$

$$\hat{\mathbf{x}} = LN(\mathbf{z} + \psi(\mathbf{z})). \tag{9}$$

where $\psi$ is a two-layer MLP and $\hat{\mathbf{x}}$ is the knowledge-embedded input of the graph neural network in the second iteration. In practice, we only implement two iterations considering the time consumption since the classification performance can not be significantly improved over more iterations.

**Loss Function Design.** We use the focal loss for object and predicate classification [14, 29] in both meta-embedding learning and scene graph prediction tasks due to the data distribution imbalance. The focal loss we used is formulated as in [29]:

$$\mathcal{L}_{focal} = \alpha(1-p)^{\gamma}\log(p), \tag{10}$$

where $p$ is the logits of a prediction of object or predicate, $\alpha$ is the normalized inverse frequency for object classification and an edge/no-edge weight for the predicates, and the $\gamma$ is a hyper-parameter.

We follow the parameter settings in [29]. To enforce the meta-embedding close to the class latent vectors in metric space, we adopt the Euclidean distance $d$ between the output of the graph encoder and the corresponding meta-embedding as a loss:

$$\mathcal{L}_{dist} = \sum_{v \in \mathcal{V}} d(\mathbf{h}_v, \mathbf{e}_{meta}^v) + \sum_{e \in \mathcal{E}} d(\mathbf{h}_e, \mathbf{e}_{meta}^e), \tag{11}$$

where $\mathbf{e}_{meta}^v$ and $\mathbf{e}_{meta}^e$ are the corresponding meta-embedding of the categories of node $v$ and edge $e$. In the meta-embedding learning task, we formulate the loss function as:

$$\mathcal{L}_{meta} = \mathcal{L}_{focal}^{obj} + \mathcal{L}_{focal}^{pred} + \mathcal{L}_{dist}. \tag{12}$$

Empirically, we use the summation of the three components with the same weight. In the scene graph prediction task, we only use the focal loss for object and predicate classification.

$$\mathcal{L}_{sg} = \lambda \mathcal{L}_{focal}^{obj} + \mathcal{L}_{focal}^{pred}, \tag{13}$$

where $\lambda$ is a hyper-parameter to re-weight the components as in [29].

## 4   Experiments and Evaluations

**Dataset and Task Description.** We train the meta-learning auto-encoder and the scene graph prediction model on the 3DSSG dataset [29] [1], a 3D scene graph dataset based on 3RScan [28]. The dataset annotates support, proximity, and comparative relationships among daily indoor objects. With the same sub-scene split in [29], there are 3582 scenes in the training set and 548 for evaluation. On average, there are 9 objects per scene of 160 categories with 27 kinds of relationships, including none relation. We evaluate our model and compare it to others in two standard tasks proposed in [32]. (1) The scene graph classification (SGCls) scenario is aimed to predict both object and predicate semantic classes. (2) In the predicate classification (PredCls) task, the model only needs to predict the predicate categories with the ground truth labels of object entities.

**Implementation Details.** Our model is implemented in PyTorch. We trained our model on an Nvidia RTX 2080Ti GPU in a personal computer platform for 40 epochs with the ADAM optimizer. The initial learning rate is set to 0.0001, and the decay rate is 0.7 for every ten epochs. We trained the meta-embedding learning network and the scene graph prediction network separately. We followed the focal loss parameter settings in [29]. In the scene graph prediction model, we used pre-trained multi-scale PointNets to encode the point cloud. The multi-scale PointNet is also trained on the 3DSSG dataset. We replaced the faster-RCNN [19] image encoder with the same pre-trained PointNet used in our model to compare with image-based methods [4, 22, 29] [2]. We selected the meta-embedding vectors according to the five most confident class predictions in the knowledge fusion stage. In the PredCls task, we assigned the corresponding object meta-embedding as the input of the graph encoder since the ground truth of objects is known. We trained each model three times to calculate the standard deviation. All models are trained on the 3DSSG dataset with the same random seeds and the same split for a fair comparison.

**Metrics.** Conventionally, we compute the recall at the top-n (R@n) triplets predicted by each model. The triplet is considered correct when the subject, predicate, and object are all valid. As defined in [15], the recall is the fraction of the correct top-n triplets against the ground truth. Additionally, since the long-tail phenomenon in the dataset annotations, we also adopt mean recalls (mR@n) [4, 26] to evaluate the performance on the unevenly sampled relations. The mean recall is computed as the average of the recalls on each predicate class.

**Comparison with Related Methods.** We introduce several baselines to compare with our proposed method. *Co-Occurrence*: we propose a relation prediction baseline with only co-occurrence statistics of the training set. This baseline outputs the most frequent predicate given the subject and object categories predicted by the multi-scale PointNet. This simple model draws a scratch line for

---

[1]We used the 3DSSG dataset [29] under the permission of its creators and authors. We contacted the author by email (johanna.wald@tum.de).

[2]We implemented the KERN model based on its released code (`https://github.com/yuweihao/KERN`) with MIT license. The SGPN and the Sechemata models have no public code for now. We reproduced them based on their papers.

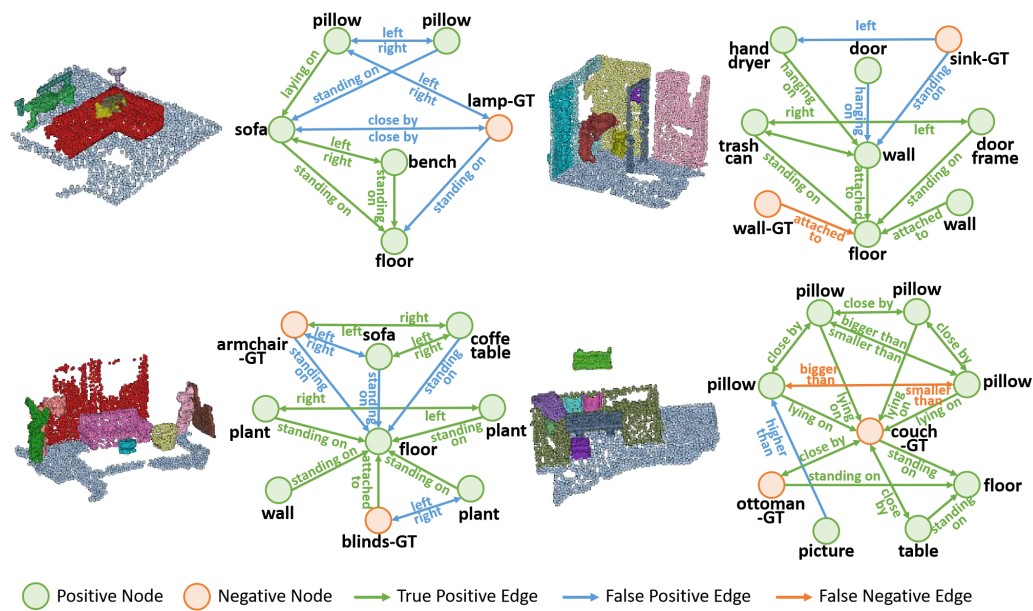

Figure 4: Examples of predicted scene graphs from our proposed method. The green nodes are correctly classified by our model. The orange nodes are false negatives. The green edge results are true positives at the R@50 setting. The blue edges are correct in commonsense but not annotated in the ground truth. The orange edges are the annotated relationships missed by our model.

knowledge-based methods on the evaluated dataset. Since this model is deterministic, the standard deviation is zero for both SGCls and PredCls tasks. *KERN*: the knowledge-embedded routing network is a robust statistical knowledge-based method that incorporates the co-occurrence probability into the scene graph prediction [4]. *SGPN*: the scene graph prediction network is proposed with the 3DSSG dataset, which is the first scene graph prediction model evaluated on this dataset [29]. *Schemata*: this baseline is a recently proposed scene graph prediction model with learning-based knowledge. It encodes the class-dependent prototypes from the previous decoded perceptual results [22], which is a similar approach to ours.

Table 1: Quantitative results of the evaluated methods in SGCls and PredCls tasks in recall, with and without graph constraint (GC). We additionally report our method with and without the meta-embedding (ME) intervention. The standard deviation is reported after the mean recall value.

|  |  | | SGCls | | | PredCls | |
| --- | --- | --- | --- | --- | --- | --- | --- |
|  | Model | R@20 | R@50 | R@100 | R@20 | R@50 | R@100 |
| w/ GC | Co-Occurrence | $0.148_{\pm0.000}$ | $0.197_{\pm0.000}$ | $0.199_{\pm0.000}$ | $0.347_{\pm0.000}$ | $0.474_{\pm0.000}$ | $0.479_{\pm0.000}$ |
|  | KERN [4] | $0.203_{\pm0.007}$ | $0.224_{\pm0.008}$ | $0.227_{\pm0.008}$ | $0.468_{\pm0.004}$ | $0.557_{\pm0.007}$ | $0.565_{\pm0.007}$ |
|  | SGPN [29] | $0.270_{\pm0.001}$ | $0.288_{\pm0.001}$ | $0.290_{\pm0.001}$ | $0.519_{\pm0.004}$ | $0.580_{\pm0.005}$ | $0.585_{\pm0.004}$ |
|  | Schemata [22] | $0.274_{\pm0.003}$ | $0.292_{\pm0.004}$ | $0.294_{\pm0.004}$ | $0.487_{\pm0.004}$ | $0.582_{\pm0.007}$ | $0.591_{\pm0.006}$ |
|  | Ours (w/o ME) | $0.282_{\pm0.002}$ | $0.299_{\pm0.001}$ | $0.301_{\pm0.001}$ | $0.529_{\pm0.004}$ | $0.592_{\pm0.004}$ | $0.598_{\pm0.005}$ |
|  | **Ours** | $\mathbf{0.285}_{\pm0.001}$ | $\mathbf{0.300}_{\pm0.001}$ | $\mathbf{0.301}_{\pm0.001}$ | $\mathbf{0.593}_{\pm0.004}$ | $\mathbf{0.650}_{\pm0.004}$ | $\mathbf{0.653}_{\pm0.004}$ |
| w/o GC | Co-Occurrence | $0.141_{\pm0.000}$ | $0.202_{\pm0.000}$ | $0.258_{\pm0.000}$ | $0.351_{\pm0.000}$ | $0.556_{\pm0.000}$ | $0.706_{\pm0.000}$ |
|  | KERN [4] | $0.208_{\pm0.007}$ | $0.247_{\pm0.007}$ | $0.276_{\pm0.005}$ | $0.483_{\pm0.003}$ | $0.648_{\pm0.006}$ | $0.772_{\pm0.011}$ |
|  | SGPN [29] | $0.282_{\pm0.001}$ | $0.326_{\pm0.001}$ | $0.353_{\pm0.001}$ | $0.545_{\pm0.006}$ | $0.701_{\pm0.001}$ | $0.824_{\pm0.002}$ |
|  | Schemata [22] | $0.288_{\pm0.001}$ | $0.335_{\pm0.003}$ | $0.363_{\pm0.002}$ | $0.496_{\pm0.002}$ | $0.671_{\pm0.003}$ | $0.802_{\pm0.009}$ |
|  | Ours (w/o ME) | $0.293_{\pm0.001}$ | $0.338_{\pm0.003}$ | $0.367_{\pm0.003}$ | $0.549_{\pm0.004}$ | $0.716_{\pm0.005}$ | $0.824_{\pm0.008}$ |
|  | **Ours** | $\mathbf{0.298}_{\pm0.002}$ | $\mathbf{0.343}_{\pm0.004}$ | $\mathbf{0.370}_{\pm0.002}$ | $\mathbf{0.622}_{\pm0.005}$ | $\mathbf{0.784}_{\pm0.004}$ | $\mathbf{0.883}_{\pm0.002}$ |

Table 1 shows that our model achieves the best relation recall than other approaches and the state-of-the-art performance in both SGCls and PredCls tasks. It confirms the effectiveness of our graph network design and the intervention of the meta-embedding. Additionally, we show the results

Table 2: Quantitative results of the evaluated methods in SGCls and PredCls tasks in mean recall. We additionally report our method ablated on the meta-embedding (ME) intervention. The standard deviation is reported after the mean recall value.

| | SGCls | | | PredCls | | |
|---|---|---|---|---|---|---|
| Model | mR@20 | mR@50 | mR@100 | mR@20 | mR@50 | mR@100 |
| Co-Occurrence | $0.088_{\pm 0.000}$ | $0.127_{\pm 0.000}$ | $0.129_{\pm 0.000}$ | $0.338_{\pm 0.000}$ | $0.474_{\pm 0.000}$ | $0.479_{\pm 0.000}$ |
| KERN [4] | $0.095_{\pm 0.011}$ | $0.115_{\pm 0.012}$ | $0.119_{\pm 0.009}$ | $0.188_{\pm 0.007}$ | $0.256_{\pm 0.010}$ | $0.265_{\pm 0.009}$ |
| SGPN [29] | $0.197_{\pm 0.001}$ | $0.226_{\pm 0.006}$ | $0.231_{\pm 0.005}$ | $0.321_{\pm 0.004}$ | $0.384_{\pm 0.006}$ | $0.389_{\pm 0.006}$ |
| Schemata [22] | $0.238_{\pm 0.012}$ | $0.270_{\pm 0.002}$ | $0.272_{\pm 0.002}$ | $0.352_{\pm 0.008}$ | $0.426_{\pm 0.005}$ | $0.433_{\pm 0.005}$ |
| Ours (w/o ME) | $0.219_{\pm 0.004}$ | $0.242_{\pm 0.004}$ | $0.245_{\pm 0.005}$ | $0.353_{\pm 0.011}$ | $0.410_{\pm 0.007}$ | $0.415_{\pm 0.010}$ |
| **Ours** | $\mathbf{0.244}_{\pm 0.011}$ | $\mathbf{0.286}_{\pm 0.008}$ | $\mathbf{0.288}_{\pm 0.007}$ | $\mathbf{0.566}_{\pm 0.011}$ | $\mathbf{0.635}_{\pm 0.001}$ | $\mathbf{0.638}_{\pm 0.001}$ |

evaluated in mean recall in Table 2. Our model can still achieve the best performance. Our model improves the mR@100 of 4.3% and 22.3% in SGCls and PredCls tasks with the meta-embedding compared to our ablated version without learned knowledge. This comparison proves that our meta-embedding can improve the predictions in marginally sampled predicate categories. We also illustrate several examples of predicted scene graphs in Figure 4. We provide more examples in the supplementary material.

Compared with the Schemata method, our model performs better in relationship prediction tasks. The reason behind the difference could be that our meta-embedding is encoded from a more distinguishable latent space than the schemata, as shown in Figure 5. The visualized results confirm that the knowledge learned from the class information is more representative than that from visual information with inevitable perceptual confusion.

However, there is a massive gap in recall values while comparing the SGCls and the PredCls tasks. The performance of relationship prediction drops gravely without the object annotations. It presents the limitation of our model in object identification. This failure is caused by the severe geometric incompleteness of point clouds and the long-tail distribution of the annotations. A potential solution is using a stronger point set encoder than the PointNet to extract robust geometric features from incomplete scans.

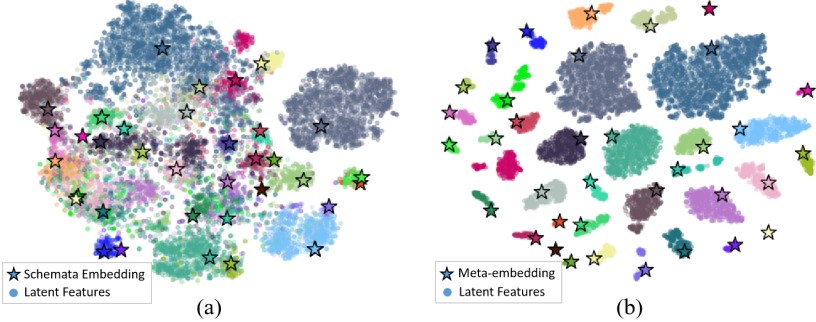

Figure 5: The t-SNE visualization [27] of the object latent space of the Schemata [22] (a) and our method (b). It shows clearly that our prototypical knowledge embedding is learned from a more distinguishable latent space than the Schemata, which extracts the knowledge from perceptual latent embedding.

**Analysis on the Meta-embedding.** We illustrate the t-SNE visualization of the output object embedding of the graph encoder in the meta-learning stage and the object meta-embedding in (b) of Figure 5. There are two points worth noticing. Firstly, the learned meta-embedding can locate at a similar area of its related class in metric space, which validates its potential to be prototypes of semantic categories. Secondly, the sub-spaces represented by the meta-embedding are distinguishable that help to enlarge the distance between categories after injecting knowledge into perception inference. In Figure 6, we visualize the latent spaces in the predicate classification task with and without the meta-embedding. It illustrates that the predicate representations are more separable after knowledge intervention.

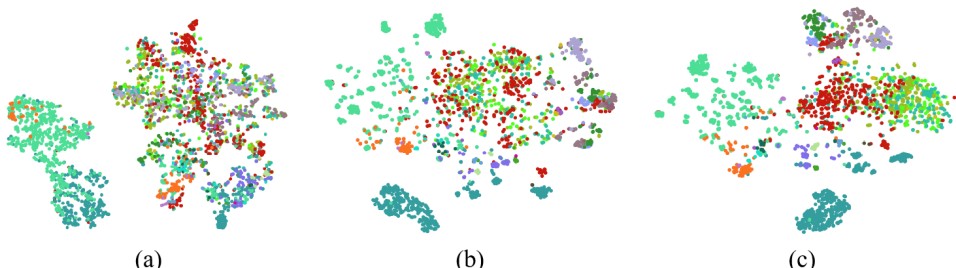

Figure 6: The t-SNE visualization of the latent spaces in the predicates classification task. The figures show the geometrical features from PointNet in (a), the contextualized features by the graph neural network in (b), and the latent space after knowledge intervention in (c).

Table 3: Quantitative results of our proposed method in SGCls and PredCls tasks on different splits of the 3DSSG dataset. We report our method ablated on the meta-embedding (ME) intervention. The percentage of the used training set is noted in parentheses. The standard deviation is reported after the mean recall value.

| | SGCls | | | PredCls | | |
|---|---|---|---|---|---|---|
| Model | mR@20 | mR@50 | mR@100 | mR@20 | mR@50 | mR@100 |
| Ours w/o ME (100%) | $0.219_{\pm0.004}$ | $0.242_{\pm0.004}$ | $0.245_{\pm0.005}$ | $0.353_{\pm0.011}$ | $0.410_{\pm0.007}$ | $0.415_{\pm0.010}$ |
| Ours w/ ME (0%) | - | - | - | $0.365_{\pm0.005}$ | $0.428_{\pm0.003}$ | $0.430_{\pm0.004}$ |
| Ours w/ ME (25%) | $0.185_{\pm0.005}$ | $0.219_{\pm0.004}$ | $0.220_{\pm0.005}$ | $0.505_{\pm0.009}$ | $0.557_{\pm0.004}$ | $0.559_{\pm0.004}$ |
| Ours w/ ME (50%) | $0.227_{\pm0.001}$ | $0.249_{\pm0.001}$ | $0.252_{\pm0.001}$ | $0.557_{\pm0.005}$ | $0.600_{\pm0.005}$ | $0.603_{\pm0.005}$ |
| **Ours w/ ME (100%)** | $\mathbf{0.244}_{\pm0.011}$ | $\mathbf{0.286}_{\pm0.008}$ | $\mathbf{0.288}_{\pm0.007}$ | $\mathbf{0.566}_{\pm0.011}$ | $\mathbf{0.635}_{\pm0.001}$ | $\mathbf{0.638}_{\pm0.001}$ |

Additionally, to understand the importance of the meta-embedding intervention, we train the scene graph classification model with more minor splits. The results are reported in Table 3. For the SGCls task, our model can achieve competitive results with the backbone model with only 50% of the training set with the meta-embedding. For the PredCls task, our model can even give reasonable predicate classification results without any perception information with the object labels and corresponding meta-embedding. This attempt could also be extended for scene generation applications. For example, our model can output a plausible scene organization with only graph inference based on the learned meta-embedding given object numbers and categories. Another interesting observation is that our design preserves the possibility of training the meta-embedding from other data sources, which could be larger than the training set used in the scene graph prediction model. It is also the reason that we maintain the independence of the two modules instead of training the whole model in an end-to-end manner.

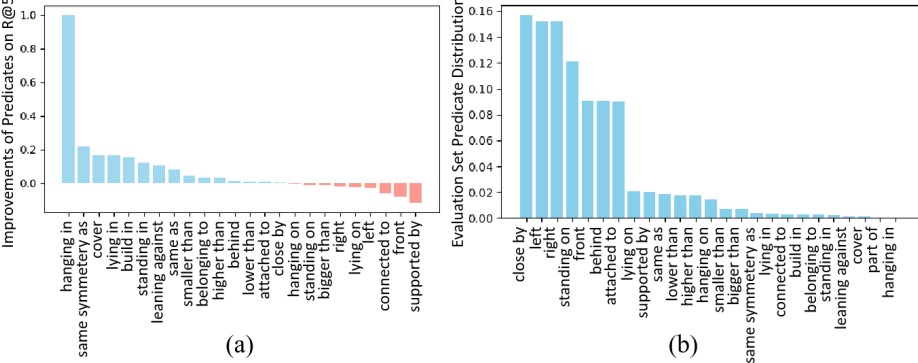

Figure 7: The relative improvement of predicate recalls on R@5 with the intervention of the meta-embedding(a) and the predicates statistical distributions (b). The meta-embedding intervention enhances the predictions especially on rarely sampled predicate categories.

Furthermore, we illustrate the improvements of predicate predictions on R@5 with the meta-embedding intervention and the distribution of each predicate category in the evaluation set in Figure 7. We can see that the meta-embedding intervention can notably improve the recall of the less frequently sampled predicates. However, our model could damage the performance of predicting categories with considerable frequency. Since the fused meta-embedding could introduce extra information of the most related categories, it could be noises for the most frequent classes but guidance for rare-sampled ones. Fortunately, the recalls of the frequent predicates are qualified enough because of their large proportion in the training set. We provide more quantitative results of the predicate predictions in the supplementary material.

## 5    Conclusion, Discussion, and Future Work

We proposed a novel knowledge-inspired 3D scene graph prediction method in this paper. We decomposed our final destination into two sub-tasks. For the first task of knowledge embedding learning, we designed a graph auto-encoder to extract class-dependent prototypical knowledge solely based on semantic category information without adverse effect of perceptual confusion. Secondly, we incorporated the learned knowledge embedding in the scene graph prediction task for an accurate object and predicate classification performance. Comprehensive experiments verified that our method could extract and inject the prior knowledge to achieve more accurate relationship prediction than the existing approaches pertaining to this subject.

However, our method still has several limitations yet to overcome. First, our multi-scale point set encoder could fail to predict confident initial class guesses when experiencing severe incompleteness. The inaccurate object identification impairs the relation predictions. Second, our knowledge intervention could not help the predictions for the most frequent semantic categories since the chosen knowledge might be noises for the inference of those classes. Moreover, our prior learned knowledge is highly dependent on the variation and the completeness of the training set. Considering the personal information security, the data used in our paper contains only uncolored scene scatter scans without any personally identifiable information nor any sensitive content. However, for the potential negative societal impacts, in real-world robot navigation tasks, inaccurate prediction of object entities and their relationships could lead agents to the wrong target and raise human safety concerns. To avoid this potential problem, we suggest proposing a security protocol in case of dysfunction of our scene understanding algorithm in real-world robotic applications.

Our near-term research efforts are geared towards immediate improvement with better solutions. Other aspects of 3D scene understanding and applications could be researched in the long term. For example, we could formulate the dynamic scene graph generation task to capture spatio-temporal cues from other data modalities (e.g., videos). Additionally, it would be of interest to researchers to explore the registration of virtual agents in dynamic scene graphs and enhance interactions with the real world in augmented reality (AR) applications.

## Acknowledgements

This research is supported in part by National Key R&D Program of China (No.2018YFB1700603), National Natural Science Foundation of China (No.61672077 and 61532002), Beijing Natural Science Foundation-Haidian Primitive Innovation Join Fund (L182016), and National Science Foundation of USA: IIS-1812606 and IIS-1715985.

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
