# Supplementary Material

**Submission 9603**

## A    Data Preparation

In this section, we introduce the data preparation for the PointNet and scene graph prediction model training. We generate point clouds from the vertex of scene meshes. For each object instance of a scene, we sample the point number to 1024 by the farthest point sampling (FPS) algorithm. The point cloud of each object and object pair are regulized in a box of [-0.5, 0.5] with the center in the origin point. For the object classification, we adopt random rotations along axis z to enhance the generalizability of our model. However, since the proximity relationships (e.g., left and right) are sensitive to the orientation of the object pair, we abandon the rotation augmentation in the predicate classification task.

## B    Scene Graph Prediction

**Additional Implementation Details.** We use a multi-scale version of PointNet [1] as our object and predicate initial encoders. In detail, we sample the point set into three sub-sets with 1024, 258, and 128 points. For each perception scale, we utilize the original PointNet model to extract geometric features. Then, we concatenate the features and transform the vector using another three-layer feed-forward network. We organize the object point clouds in one batch during the training process. The learning rate of training the multi-scale PointNet is set to 0.0001, and the decay rate is 0.7 for every ten epochs. We train the multi-scale PointNet for 100 epochs in object classification task and 40 epochs for predicate classification with the focal loss [2] mentioned in our paper.

**Predicate Classification Results.** We report additional quantitative predicate classification results in Table 1 on R@5. With the meta-embedding, our model can achieve more accurate predicate classification, especially in marginally sampled relationships. Though the intervention of the meta-embedding could reduce the prediction recall of some relationships, the prediction results of those categories are still relatively high compared to other classes.

Table 1: Quantitative results of the predicate classification on R@5. We report our method ablated on the meta-embedding (ME) intervention.

| Relationships | Ours w/o ME | Ours w/ ME | Relationships | Ours w/o ME | Ours w/ ME |
|---|---|---|---|---|---|
| supported by | **0.806** | 0.692 | standing on | **0.995** | 0.986 |
| left | **0.911** | 0.881 | lying on | **0.970** | 0.948 |
| right | **0.908** | 0.889 | hanging on | **0.987** | 0.981 |
| front | **0.750** | 0.670 | connected to | **0.794** | 0.735 |
| behind | 0.656 | **0.668** | leaning against | 0.368 | **0.474** |
| close by | 0.898 | **0.901** | part of | 0.833 | **0.833** |
| bigger than | **0.741** | 0.729 | belonging to | 0.645 | **0.677** |
| smaller than | 0.682 | **0.729** | build in | 0.788 | **0.939** |
| higher than | 0.836 | **0.867** | standing in | 0.680 | **0.800** |
| lower than | 0.856 | **0.867** | cover | 0.444 | **0.611** |
| same symmetry as | 0.260 | **0.480** | lying in | 0.278 | **0.444** |
| same as | 0.495 | **0.579** | hanging in | 0.000 | **0.999** |
| attached to | 0.986 | **0.994** | | | |

35th Conference on Neural Information Processing Systems (NeurIPS 2021).

**Scene Graph Prediction Examples.** We show more examples of predicted scene graphs from our proposed method in Figure 1. The green nodes are correctly classified objects. The orange nodes are false results. The green edge represents true predicate prediction at the R@50 setting. The blue edges are correct results in commonsense but not annotated in the ground truth. The orange edges are the annotated relationships missed by our model.

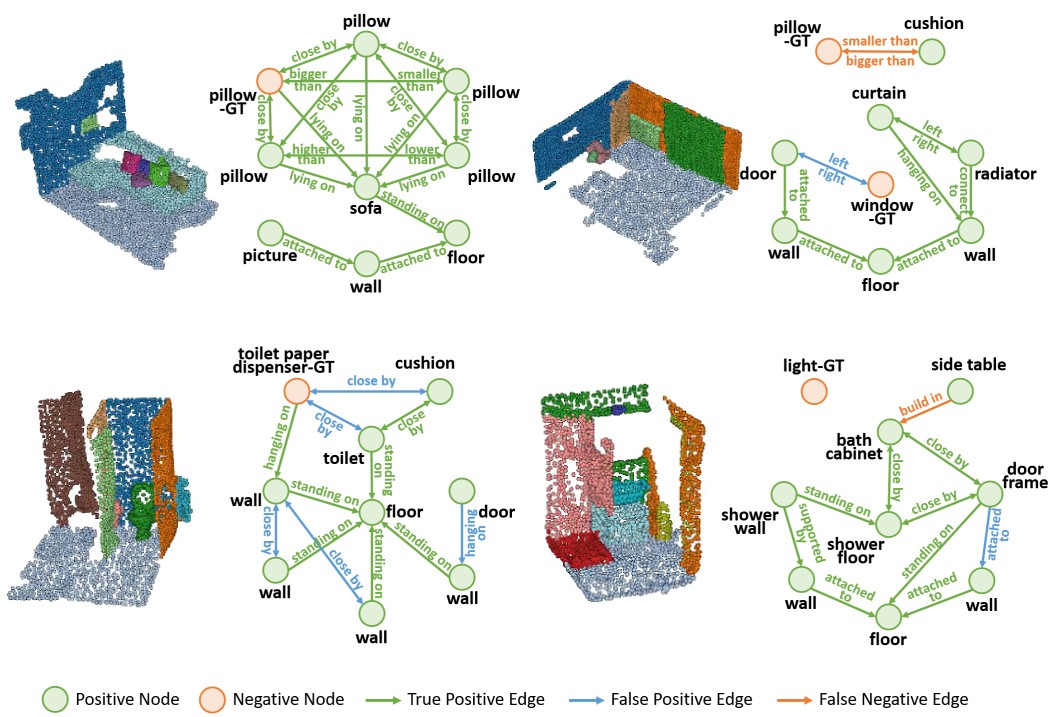

Figure 1: Examples of predicted scene graphs from our proposed method.