# OpenReview forum: "Knowledge-inspired 3D Scene Graph Prediction in Point Cloud"
_NeurIPS.cc/2021/Conference — NeurIPS 2021 Poster_

### Official Review · Reviewer_YwYy · 2021-07-16

**Rating:** 5
**Confidence:** 4

**Summary:**

In this paper, the authors study the 3D scene graph prediction task. The major contribution is the proposed “object/predicate meta-embedding,” which learns class-specific embedding to assist scene graph prediction. Ablation study verifies the effectiveness of the proposed embedding learning method.

**Limitations And Societal Impact:**

The limitations and societal impacts are discussed in the conclusion section.

**Main Review:**

Strengths:
1. The additional object and predicate embedding improves the scene graph prediction performance.

2. The analyses and feature visualization show that the meta-embedding does learn class-specific representation as desired.

Weaknesses:
1. In the claimed contribution (Line 53), it is unclear what does the first point (“we introduce a novel ... method that directly functions on 3D scene point clouds”) refers to. The framework seems to have the same input-output as the previous 3D scene graph prediction studies [29].

2. The “Knowledge-inspired Scene Graph Prediction Model” section and Figure 3 are confusing. The two-round approach (the loop from object prediction to initial feature) is barely introduced in the text, and thus makes the inference pipeline difficult to understand.

    A question is that whether more than two rounds with more accurate object/predicate prediction further improves the performance.

3. In addition to the reported metrics in image scene graph prediction, it might be helpful to also benchmark the studies following previous 3D scene graph studies’ metrics [29].

4. a. The term “commonsense” could be misleading. The learned “object/predicate meta-embedding” is more like external class-specific memories, instead of “commonsense” or “knowledge.”

5. It would be helpful to conduct additional ablation studies and analyses on the meta-embedding design. For example, what if keeping the meta-embedding but learn it E2E instead of using the auto-encoder-like structure.

6. Despite the performance improvement, the extra embedding also increases the training and inference cost, as it requires two inference rounds and storing additional memory-related parameters.

7. The proposed method seems generalizable to image scene graph prediction. It would be helpful to also benchmark the study on image scene graph prediction, or show the unique design for the 3D task.


**Time Spent Reviewing:**

3

---

> ### Author Response · Authors · 2021-08-09
> **Replies to Reviewer YwYy**
>
> Thanks for the comments and constructive suggestions.
>
> (Q1) “In the claimed contribution (Line 53), it is unclear what does the first point refers to. ... seems to have the same input-output as [29].”
>
> (A1) We wish to emphasize a novel knowledge-inspired approach in this paper (serving as our first contribution). This paper re-formulates the problem to design our knowledge-based method in order to substantiate this claim. Although it has the same input-outputs as [29], a knowledge-intervened approach in 3D qualifies as a contribution, which is echoed by [Reviewer rms6] who wrote "the idea to learn knowledge solely based on class labels and their regular graphic structures is novel." [Reviewer r35L] also wrote "the contributions are highlighted clearly."
>
> (Q2) “The two-round approach (the loop from object prediction to initial feature) is barely introduced ... makes the inference pipeline difficult to understand... whether more than two rounds with more accurate object/predicate prediction further improves the performance.”
>
> (A2) We plan to make a clearer statement in the final camera-ready version. In the second iteration (the loop from object prediction to initial feature), the selected meta-embedding is fused with the initial feature based on Eq. 8 and Eq. 9, which corresponds to the Add&Norm operation in Fig 3. The codes in our supplementary files document all technical details, which were already examined and verified by [Reviewer r35L]. Since the second question is also raised by another reviewer, please refer to our answer in the Q2 of [Reviewer 8qYe].
>
> (Q3) “In addition to the reported metrics in image scene graph prediction, it might be helpful to also benchmark the studies following previous 3D scene graph studies' metrics [29].”
>
> (A3) We must explicitly mention the problem associated with the metrics referred to by the paper [29]. The paper [29] claimed that the metrics they used are the ones proposed in [a], but their metrics (actually employed) were different from [a] after we checked their evaluation codes which were made available to us. To address such a concern on possible inconsistency and also try to enhance the overall reliability of our experiment results with more confidence, we decided to adopt the metrics proposed in [a] to evaluate our model and all the related methods, which was also done in previous related scene graph prediction methods [b, c, d].
>
> (Q4) “... The learned "object/predicate meta-embedding" is more like external class-specific memories, instead of "commonsense" or "knowledge."”
>
> (A4) We did not mean that the commonsense is literally equivalent to the meta-embedding in our paper. Our main idea is to encode the regular patterns between object pairs into a set of class-related embedding. The previous work [d] also adopted class-dependent representations as the carriers of relationship knowledge, resulting in and confirming a similar conclusion. We believe that we have clearly introduced the relation between knowledge and the meta-embedding, as [Reviewer rms6] wrote that our work might “brings some new insights to other knowledge-inspired tasks.”
>
> (Q5) “It would be helpful to conduct additional ablation studies and analyses on the meta-embedding design. ... but learn it E2E instead of using the auto-encoder-like structure.”
>
> (A5) We shall add more analysis in our final camera-ready version. We tried the E2E structure and achieved a competitive performance with a full training set. Nonetheless, an E2E structure forces different modules to be trained on a shared training set, definitely confining the domain and the expandability of the knowledge learning model. In contrast, our design preserves the possibility of training the meta-embedding from more data samples, which could be larger than the training set used in the scene graph prediction model.
>
> (Q6) “Despite the performance improvement, the extra embedding also increases the training and inference cost. ...”
>
> (A6) It may be noted that, we must deal with the compromise between performance improvement and model complexity. Although it needs extra inference time and memory cost, our model costs only 23s to process 548 scenes during the testing stage (on a Nvidia RTX 2080Ti GPU), and the meta-embedding only occupies 376 KB in the npy format. The extra cost is obviously acceptable for an economical computation platform.
>
> (Q7) “The proposed method seems generalizable to image scene graph prediction. ..., or show the unique design for the 3D task.”
>
> (A7) We must say that a generalized framework for multimodal inputs is not the goal of this research work, even though our primary idea has the potential to extend to 2D scene graph inference tasks. We adopted the multi-scale PointNet as a part of the encoder, which is a unique design for 3D and limits our current network design only to taking 3D point cloud data as input. It is inevitable to modify our encoder in order to process image data. We very much appreciate your constructive suggestion, which will provide guidance for our future efforts.
>
> [a] C. Lu, R. Krishna, M. S. Bernstein, and F. Li. Visual relationship detection with language priors. In Springer ECCV, volume 9905 of Lecture Notes in Computer Science, pages 852–869, 2016.
>
> [b] T. Chen, W. Yu, R. Chen, and L. Lin. Knowledge-embedded routing network for scene graph generation. In IEEE CVPR, pages 6163–6171, 2019.
>
> [c] J. Gu, H. Zhao, Z. Lin, S. Li, J. Cai, and M. Ling. Scene graph generation with external knowledge and image reconstruction. In IEEE CVPR, pages 1969–1978, 2019.
>
> [d] S. Sharifzadeh, S. M. Baharlou, and V. Tresp. Classification by attention: Scene graph classification with prior knowledge. CoRR, abs/2011.10084, 2020.

---

> > ### Comment · Reviewer_YwYy · 2021-08-26
> > **Post Rebuttal**
> >
> > Thank you for your response.
> >
> > 1. Can I interpret the "knowledge" as a learned object (predicate)-class-specific representation/memory? If so, my understanding of the contribution is introducing such "class-dependent representation" to the 3D SG task?
> >
> > 2, 4. Thank you, related discussions and clarifications are important in the updated version. And as a reader, I personally find "commonsense" somewhat misleading
> >
> > 5. The concern behind this is to what degree the proposed class-specific representation is working, instead of better performance learned achieved with extra parameters.
> >
> > 6. Thank you for the information. Related discussion might be necessary as this is a straightforward concern when iteration is involved.
> >
> > 7. The concern behind this is that the proposed class-specific representation is not unique for the 3D task, seems that we can replace the encoder from "multi-scale PointNet" with a CNN to have it work on images? The contribution would be stronger if I missed any 3D-specific design in this "class-specific representation."
> >
> > Thank you.

---

> > > ### Author Response · Authors · 2021-08-28
> > > **Replies to the Post Rebuttal**
> > >
> > > Thank you for your responses, and now we wish to provide further discussion below in order to address your current concerns towards a better understanding. We appreciate your kindness.
> > >
> > > (Q1) Can I interpret the "knowledge" as a learned object (predicate)-class-specific representation/memory? If so, my understanding of the contribution is introducing such "class-dependent representation" to the 3D SG task?
> > >
> > > (A1) We agree with the understanding mentioned above. Since knowledge can be represented in various forms (e.g., vectors, graphs, sentences), we chose to encode the semantics-critical structural patterns as knowledge into a set of class-specific embedding, which is expected to contribute to the knowledge-based 3D scene graph inference task.
> > >
> > > (Q2, 4) Thank you, related discussions and clarifications are important in the updated version. And as a reader, I personally find "commonsense" somewhat misleading.
> > >
> > > (A2, 4) We suppose that you might mean that the word “commonsense” tends to be a natural cognitive ability instead of regular spatial patterns in practice, considering the definition in Dictionary Collins: ‘commonsense’ is the natural ability to make good judgments and behaves in a practical sensible way. We will reconsider and choose a more appropriate term for the regular patterns in our updated version in order to prevent potential confusion about the word “commonsense” for general readers.
> > >
> > > (Q5) The concern behind this is to what degree the proposed class-specific representation is working, instead of better performance learned achieved with extra parameters.
> > >
> > > (A5) We suppose you might mean that, to what extent the meta-embedding can be effective. Especially considering your earlier-mentioned question Q5 (in your review), you might be interested in the class-specific representation encoded by different models. However, we only tested the auto-encoder and E2E structure, and we have made careful conclusions based on the existing experimental results. To address the performance limits of our proposed class-specific embedding (especially in the general capacity of network architecture and design beyond the traditional boundary of the tested cases above), it would require much more comprehensive study on the challenging issues (pertinent to network architecture and design), with/without various GNN mechanisms (e.g., message encoding, pooling methods) and learning strategies (e.g., joint learning, few-shot learning, separate models) to push the learned embedding towards possible limit cases (e.g., more imbalanced samples than the used dataset). This undertaking is definitely valuable, but it would take tremendous amount of work and might not be validated in the updated version (given the time pressure). This valuable question provides an impetus for us to explore the effectiveness of general network architecture and design involving knowledge embedding in our future efforts.
> > >
> > > (Q6) Thank you for the information. Related discussion might be necessary as this is a straightforward concern when iteration is involved.
> > >
> > > (A6) We will add more related discussions in our final version.
> > >
> > > (Q7) The concern behind this is that the proposed class-specific representation is not unique for the 3D task, seems that we can replace the encoder from "multi-scale PointNet" with a CNN to have it work on images? The contribution would be stronger if I missed any 3D-specific design in this "class-specific representation."
> > >
> > > (A7) Indeed, the knowledge learning model could be applied to the 2D benchmarks by modifying the amount of the meta-embedding compatible with the scale of semantic classes. However, we only discussed and validated our contributions in the 3D task in the current paper, since we did not test our model in the 2D domain. The suggested extension would need more effort on the study of our knowledge encoding method in conjunction with different relationship types and sample distributions. The current work only represents one of the initial attempts towards knowledge encoding from non-visual information in 3D task. Your suggestion is constructive for our future efforts to involve extra 3D-specific features (e.g., relative positions and scales) pertinent to the 3D-specific task. It is promising to explore new, generic, (and compatible) knowledge learning models for various domains.

---

> > > > ### Comment · Reviewer_YwYy · 2021-09-02
> > > > **Replies**
> > > >
> > > > Thank you for the discussion. Your responses help me better understand the work.
> > > >
> > > > I agree with the contribution of "introducing class-specific representation to the specific task of 3D SG," and the experiments show that it works on the 3D SG task.  I'll update the rating to more positive and agree to accept the paper, with the expectation that the authors will update the potential misleading claims (e.g., the original weaknesses 1,4), and include related discussions as promised.

---

> > > > > ### Author Response · Authors · 2021-09-03
> > > > > **Re: Replies**
> > > > >
> > > > > Thank you for updating the rating point! We appreciate your kindness, and we will make improvements mentioned in the discusstions in the final updated version.

---

### Official Review · Reviewer_rms6 · 2021-07-16

**Rating:** 7
**Confidence:** 3

**Summary:**

This paper proposes a knowledge-inspired method for 3D scene graph prediction on point cloud. The key is to decompose the problem into two sub-tasks. First, a graph auto-encoder called meta-embedding is introduced to learn class-dependent prototypical representations only from the class labels to avoid visual confusions. Then a scene graph prediction model selects related meta-embedding from the first task as prior knowledge to predict credible relation triplets of 3D scene graph. Benefited from the learned commonsense knowledge, the scene graph prediction model achieves good performance on 3DSSG dataset.

**Limitations And Societal Impact:**

The societal impacts are fully discussed.
For the limitations of the method, the reviewer hope to know the robustness of the proposed method to the perturbation such as rotation, translation and scaling. The robustness to the perturbation on 3D point clouds is very important in real-world applications. Since the proposed method learn the knowledge solely from labels, it may be robust to the real-world perturbations, which is expected to be discussed in the rebuttal.

**Main Review:**

Strong points:
1. The idea to learn knowledge solely based on class labels and their regular graphic structures is novel, which avoids the visual confusion introduced by the perceptual errors in previous works. It may bring some insights to other knowledge-inspired tasks.
2. The experiments and visualizations are thoroughly provided, while the weaknesses are also discussed.
3. The paper is well-written and easy to follow.

Weak points:
1. Line 244-249 argues that the limitation of the model in object identification can be possibly solved by using a stronger point set encoder than PointNet. However, Table 1 shows that the method w/o ME can also get a good result (using PointNet), where the problem is that the performance gain in SGCLs task is not enough to fully prove the effectiveness of ME. This is expected to be discussed in the rebuttal.
2. In Line 268, the performance on frequent categories is damaged by the model. However, the reason to cause this problem is not fully discussed.
3. Line 167 states that the five most confident embeddings are selected, will the number of the embeddings influence the result? Since the selection of prior knowledge is also the key in to utilize the knowledge, this detail is expected to be discussed.
4. Figure 6 of Schemata [1] clearly confirms that object representations will get into more separable clusters after injecting their prior knowledge. Although Line 255 argues similar effects, the same visualization comparison is expected to be provided.

**Time Spent Reviewing:**

6

---

> ### Author Response · Authors · 2021-08-09
> **Replies to Reviewer rms6**
>
> Thanks for your positive comments, constructive suggestions, and kind encouragement about our work.
>
> (Q1) “... method w/o ME can also get a good result (using PointNet), ... performance gain in the SGCls task is not enough to fully prove the effectiveness of ME. ...”
>
> (A1) It may be noted that, in the results in the PredCls task, where the exact object labels are known, the ME can effectively enhance the performance in scene graph inference. However, in the SGCls task, the initial object predictions are not accurate enough to select the proper knowledge embedding. The inaccurate knowledge selection may involve noisy features in the knowledge fusion process, which only slightly improves the performance with meta-embedding. Thus, we propose a potential solution to find a more robust point set encoder that can give more accurate initial object category prediction than the PointNet, where handling geometrical incompleteness and appearance noises is necessary.
>
> (Q2) “In Line 268, the performance on frequent categories is damaged by the model. However, the reason to cause this problem is not fully discussed”
>
> (A2) We will make more discussion about Line 268 in our final version. For the most frequent categories, our model can still learn dataset bias and give accurate predictions even without the meta-embedding. Nonetheless, since the fused meta-embedding could introduce extra information of the most related categories, it could be noises for the most frequent classes but guidance for rare-sampled ones. Thus, the most improvements occur in under-represented classes, which means our performance is beyond the simple statistical bias of the dataset. This phenomenon also occurred in another knowledge-based scene graph prediction method [a].
>
> (Q3) “..., will the number of the embeddings influence the result? ...”
>
> (A3) Because this question is also raised by another reviewer, please refer to our answer in the Q2 of [Reviewer r35L].
>
> (Q4) “... Although Line 255 argues similar effects, the same visualization comparison is expected to be provided”
>
> (A4) We shall provide such visualization comparisons in our final version.
>
> [a]S. Sharifzadeh, S. M. Baharlou, and V. Tresp. Classification by attention: Scene graph classification with prior knowledge. CoRR, abs/2011.10084, 2020.

---

> > ### Comment · Reviewer_rms6 · 2021-09-02
> > **Post-rebuttal comment**
> >
> > The rebuttal addressed all my concerns. Thanks.

---

### Official Review · Reviewer_8qYe · 2021-07-16

**Rating:** 6
**Confidence:** 3

**Summary:**

The paper provides a novel two-stage pipeline for 3D scene graph prediction tasks on point cloud data.
Novel relation and object embedding methods are proposed.
Experiment results show that the proposed pipeline gives higher performance compared with related methods.

**Limitations And Societal Impact:**

Yes

**Main Review:**

Originality:

The two-state pipeline is of great novelty.
The embedding method proposed is effective.

Clarity:

The overall idea is expressed clearly.
But details on network structures are missing or fuzzy.

In meta-embedding learning auto-encoder, it is unclear to me how the network structure is obtained.
Judging by Fig 1 and Fig 2, the input contains only one-hot embeddings.
But GNN requires a network structure to operate on.
I conjecture that the graphs come from all scene graph in the dataset.
Then in that case, the input should further include graph structure instead of only labels.

Judging by Fig 3, there is a recurrent structure in the scene graph prediction model.
It is not clear to me what will happen in the first iteration before any meta-embedding can be obtained.
Without meta-embedding, updates described by equation 8 cannot be executed.
Other questions are how many iterations are required and will the classification accuracy improve over iterations.

Quality:

The work is complete and the model is sound.
Evaluation is comprehensive and convincing.
The limitation is discussed in detail.

Significance:

3D scene understanding is an important research direction with great potential.
The performance of the model proposed in this paper surpasses a large body of previous works.
The embedding method is inspiring and could potentially be used in other similar tasks.

**Time Spent Reviewing:**

5

---

> ### Author Response · Authors · 2021-08-09
> **Replies to Reviewer 8qYe**
>
> Thanks for your positive comments and constructive suggestions. We will make clearer statements in our final camera-ready version.
>
> (Q1) “In meta-embedding learning auto-encoder, it is unclear to me how the network structure is obtained. ...GNN requires a network structure to operate on. ...”
>
> (A1) The input of the GNN contains a fully-connected graph structure, assuming that any object pair can have a relation (including no relation). The objects are treated as nodes, and the predicate is the edge connecting the corresponding node pair. The one-hot object labels and predicate labels are the initial features of the nodes and edges, respectively. For more network design details, please refer to the code in our supplementary files, which were already examined and verified by [Reviewer r35L].
>
> (Q2) “... what will happen in the first iteration before any meta-embedding can be obtained. ... how many iterations are required and will the classification accuracy improve over iterations.”
>
> (A2) In the first iteration, the operation described by Eq. 8 does not execute. Instead, the output of MS-PointNet and the Diff&MLP is directly fed into the GNN to produce the initial object and predicate prediction. Eq. 8 and Eq. 9 are functional in the second iteration to fuse the selected knowledge embedding based on the initial predictions. In addition, we have tested 2 iterations and 4 iterations in our network design comprehensively. These two experiments exhibit very similar performance in both SGCls and PredCls tasks, so we set 2 iterations in our final model when taking the extra time and memory cost into consideration.

---

> > ### Comment · Reviewer_8qYe · 2021-09-01
> > **Response to the Rebuttal**
> >
> > Thanks for your response and clarifications. I have left my score as is.

---

### Official Review · Reviewer_r35L · 2021-07-21

**Rating:** 7
**Confidence:** 3

**Summary:**

The paper proposes a scene graph prediction method for 3D point clouds. The paper uses the graph network to learn class label based embeddings. It does not use appearance information from the point clouds to avoid perceptual inconsistencies. The class label dependent prototypical embeddings are learned using a graph auto-encoder with directed edges and a message passing mechanism tuned for such edges. The scene graph prediction works by extracting embeddings from subject and object points and object pair relationship in form of an edge embedding. The embeddings (geometric, meta embeddings) are fused with an MLP. The network is trained with focal loss. The proposed method is evaluated on 3DSSS dataset and achieves state of the art results on various metrics.

**Limitations And Societal Impact:**

The authors have described the limitations adequately in Section 5. However, mentioning the constraints on the subject-object pairs and type of relationshio would help readers to extend the work in other domains.

**Main Review:**

- The main novelty of the method lies in utilizing meta embeddings from class labels and fusing them with geometric embeddings to obtain the scene graphs. The approach, in my opinion, builds upon geometry based methods on point clouds [A], where the appearance information is not necessary to learn robust descriptors. However, overall adding class label based embeddings and the fusion mechanism seems novel with reference to the existing literature.

- The paper is well written and motivated. The contributions are highlighted clearly. However, main premise of the paper is to avoid appearance noise. In order to validate this hypothesis, authors may include such features and demonstrate the change in performance.

- I was able to reproduce the resutls (with 2% variance from the reported results on various metrics) with the code provided by the authors on an NVIDIA RTX 8000, 256 GB RAM.

- How do you chose the five best meta embeddings for fusion. Is it based on the attention maps ? (Line 167). What happens if the number of embeddings is more/less ?

- What is the motivation behind using Focal Loss ? The main advantage of using focal loss is during class imbalanc. Is it due to sparse mapping between the object-subject pairs here ?

- How does the technique perform with noisy data/points say sampled at a lower resolution ?

- There are a few typos
(i) Fig 1 "Labal" -> Label
(ii) Line 169, "sapces" -> spaces

[A]: Guo, Yulan, et al. "Deep learning for 3d point clouds: A survey." IEEE transactions on pattern analysis and machine intelligence (2020).

**Time Spent Reviewing:**

12

---

> ### Author Response · Authors · 2021-08-09
> **Replies to Reviewer r35L**
>
> Thanks for your positive comments and constructive suggestions.
>
> (Q1) “..., authors may include such features and demonstrate the change in performance.”
>
> (A1) We have thoroughly evaluated our model w.r.t. another knowledge-based method (Schemata [a], which extracts knowledge with appearance noises) with the demonstrated advantage. Hence, they share a similar knowledge embedding mechanism (to a certain extent), so the experimental results documented in Table 1 can draw a comparison between pure class-related knowledge and appearance-based knowledge with noises.
>
> (Q2) “How do you choose the five best meta embeddings for fusion.” and “What happens if the number of embeddings is more/less?”
>
> (A2) We shall make more explicit statements in the final camera-ready version. At the end of the first iteration, the MLP outputs a probability map of object and predicate classification from a Softmax layer. We can obtain the five most confident category predictions as the indices to select the corresponding meta-embedding. In addition, based on our experiments, the large number of selected embedding is inevitable to introduce wrong-category knowledge. On the other hand, a small meta-embedding set might contain no proper knowledge due to the accuracy of the object and predicate prediction. In practice, we choose an appropriate number 5 as a compromise.
>
> (Q3) “What is the motivation behind using Focal Loss? ...”
>
> (A3) We adopt Focal Loss because of the class imbalance, which you have mentioned in the question. The used 3DSSG dataset shows a long-tail distribution in both object and predicate categories. In this case, the focal loss is a reasonable choice to train our model.
>
> (Q4) “How does the technique perform with noisy data/points say sampled at a lower resolution?”
>
> (A4) Since we focus on knowledge learning and knowledge-intervened scene graph inference, we only test our model at a fixed resolution. However, since we employ the PointNet as the point encoder, our model’s performance relates to the robustness of the PointNet when dealing with low-resolution inputs. We could test and improve our model with your constructive suggestion in the future.
>
> (Q5) “There are a few typos (i) Fig 1 "Labal" -> Label (ii) Line 169, "sapces" -> spaces”
>
> (A5) We shall carefully re-examine our writing and correct all typos in the final version.
>
> [a] S. Sharifzadeh, S. M. Baharlou, and V. Tresp. Classification by attention: Scene graph classification with prior knowledge. CoRR, abs/2011.10084, 2020.

---

> > ### Comment · Reviewer_r35L · 2021-09-05
> > **Satisfied with the author response**
> >
> > I thank the authors for addressing my questions. Their response answer my queries, and they have also provided insights from preliminary experimental run of a suggested setting in the review. I would like to keep my rating as is.

---

### Decision · Program_Chairs · 2021-09-27

**Decision:**

Accept (Poster)

**Comment:**

This paper proposes a method for 3D scene graph prediction from point clouds in which  a graph auto-encoder model learns prototypical representations for object categories using scene graph annotations, that are used  as prior knowledge during scene graph inference from a point cloud input.  Reviewers acknowledge the novelty of the autoencoding model proposed for learning categorical priors.
Reviewers point out that the paper is not clear regarding the input of the autoencoding graph model. They further point out that vague words like ``common sense” are not well defined and are used arbitrarily in the paper. Authors are strongly encouraged to clarify the paper writing (and corresponding figures), following reviewers’ comments.